# PARTITIONED LEARNED COUNT-MIN SKETCH

## ABSTRACT

We propose *Partitioned Learned Count-Min Sketch (PL-CMS)*, a new approach to learning augmented frequent item identification in data streams. Our method builds on the learned Count-Min Sketch (LCMS) algorithm of Hsu et al. (ICLR 2019), which combines a standard Count-Min Sketch frequency estimation data structure with a learned model, by partitioning items in the input stream into two sets. Items with sufficiently high predicted frequencies have their frequencies tracked exactly, while the remaining items, with low predicted frequencies, are placed into the Count-Min Sketch data structure.

Inspired by an approach of Vaidya et al. for learning augmented Bloom filters (ICLR 2021), our PL-CMS algorithm partitions items into different sets, based on multiple predicted frequency thresholds. Each set is handled by a separate Count-Min Sketch data structure. Unlike classic LCMS, this allows our algorithm to take advantage of the full prediction space of the learned model. We demonstrate that, given fixed partitioning thresholds, the parameters of our data structure can be efficiently optimized using a convex program. Empirically, we show that, on a variety of benchmarks, PL-CMS obtains a lower false positive rate for frequent item identification as compared to LCMS and standard Count-Min Sketch.

## 1 INTRODUCTION

Frequent item identification in massive data streams is a fundamental problem, with applications across machine learning (Medini et al., 2019), networking (Tang et al., 2020; Chabchoub et al., 2009), data mining, and databases (Beyer & Ramakrishnan, 1999; Fang et al., 1998). The goal is to identify all items that appear sufficiently often in the data stream, while using a just small amount of processing space – i.e., significantly less than is required to store an exact frequencies of every item.

Commonly, this task is formalized as the $(\epsilon, k)$-frequent items (or 'heavy-hitters') problem (Woodruff, 2016). Given a data stream of length $n$, a frequency parameter $k$, and an error tolerance $\epsilon \in (0, 1)$, to solve this problem, an algorithm should identify as heavy-hitters all items with frequency $\geq \frac{n}{k}$, but no items with frequency $< (1 - \epsilon)\frac{n}{k}$. We can relax this problem, allowing the algorithm to have a small *false positive rate (FPR)*. I.e., a small fraction of items with frequency $< (1 - \epsilon)\frac{n}{k}$ that are misidentified as heavy-hitters.

A wide variety of algorithms for the $(\epsilon, k)$-frequent items problem and its variants have been proposed. These include Misra-Gries summaries (Misra & Gries, 1982), Lossy Counting (Manku & Motwani, 2002), multi-stage Bloom filters (Estan & Varghese, 2002), Space-Saving (Metwally et al., 2005), Count-Sketch (Charikar et al., 2002) and CountSieve (Braverman et al., 2015).

One of the most popular approaches is the Count-Min Sketch (CMS) data structure (Cormode & Muthukrishnan, 2005). CMS maintains a set of $t$ tables with $m$ 'buckets' each. Each time an item appears, it is hashed using $t$ random hash functions, into one bucket in each table. Each bucket stores a counter of the total frequency of the items that have been hashed into it. In this way, the $t$ buckets that an item $x$ is hashed into (one in each table) have counters at least as large as the frequency of $x$, but potentially larger, due to hash collisions. An approximate frequency of $x$ can be obtained by taking the minimum value across the tables of the buckets that $x$ hashes into. This value is always higher than the true frequency of $x$, but ideally by only a small amount if $m$ and $t$ are large enough.

CMS yields a natural approach to solving $(\epsilon, k)$-frequent items: we simply output all elements of the stream whose approximate frequency is at least $\frac{n}{k}$. Since the CMS frequency estimates are always overestimates, we thus identify all true frequent items with frequency $\geq \frac{n}{k}$. Further, one can show

that, for any $\epsilon, \delta \in (0, 1)$, CMS with $t = O(\log(1/\delta))$ tables and $m = O(k/\epsilon)$ buckets in each table misidentifies an item with frequency $< (1-\epsilon)\frac{n}{k}$ as a heavy hitter with probability at most $\delta$. That is, the algorithm achieves expected FPR $\delta$ (Cormode & Muthukrishnan, 2005). This space complexity is optimal up to logarithmic factors for worst-case input streams (Bhattacharyya et al., 2018).

## 1.1 LEARNING AUGMENTED FREQUENT ITEM IDENTIFICATION

In recent years, significant work has focused on *learning augmented algorithms*, which use machine learning models to improve the performance of classical algorithms by adjusting their behavior with respect to the learned, underlying input distribution. (Kraska et al., 2018; Mitzenmacher, 2018; Lykouris & Vassilvitskii, 2021; Purohit et al., 2018; Roughgarden, 2021). Several works have focused in particular on learning augmented approaches to frequent item estimation (Aamand et al., 2019; Du et al., 2021; Dolera et al., 2022; Li et al., 2023).

Hsu et al. (2019) propose a *Learned Count-Min Sketch* (LCMS) algorithm, which improves the performance of CMS using a learned model. In particular, their model uses the features of an item to produce a score for that item – with higher scores meant to correspond to more frequent items. The algorithm fixes a single score threshold, and for any item in the data stream whose score is above that threshold, tracks its frequency exactly. That is, an array of 'unique buckets' is maintained, where each item with score above the threshold is assigned its own counter. Items that receive a score below the threshold (i.e., that are predicted to have low frequency) are placed in a classic Count-Min Sketch data structure, which returns frequency estimates for these items.

Hsu et al. (2019) demonstrate that on several benchmarks, their LCMS data structure can output frequency estimates with much smaller error than classic CMS, for a given space budget. Intuitively, the learned model is able to identify many of the most frequent items and avoid putting them in the CMS, leading to fewer instances of infrequent items appearing frequent due to hash collisions.

Despite the success of this approach, it leaves open a natural question: the LCMS algorithm only makes very limited use of the learned model, thresholding its output at a single value. Items with very low scores (i.e., that are very unlikely to be heavy-hitters) are treated no-differently from items with scores just below the threshold. It is natural to ask if even further performance gains can be had by using the output of the learned model in a less coarse manner.

Recently, Zhang et al. (2020) propose two other learning-based variants of CMS. The first is similar to LCMS, but instead of tracking the frequencies of predicted heavy items exactly, directly returns frequency predictions from the learned model as estimates. The second splits the input items into three classes based on the model predictions. For the items with the highest predicted frequencies, the algorithm exactly tracks frequencies. For the items in next highest group, the algorithm uses the learned model frequency estimates directly. For the items with the lowest predicted frequencies, the algorithm employs a CMS data structure. Zhang et al. (2020) show favorable performance as compared to non-learning-based variants of CMS, however their approach hinges on the ability to train a highly accurate learned model, that can be used directly for frequency estimation.

## 1.2 OUR CONTRIBUTIONS

Our main contribution is *Partitioned Learned Count-Min Sketch (PL-CMS)*, an extension of LCMS that takes advantage of the full model output space, does not require a highly accurate model, and is able to achieve significantly better frequent item identification performance on several benchmarks.

The idea of PL-CMS is simple: we partition the input items into several classes, based on multiple thresholds for the learned model output score. As in LCMS, those items with the highest predicted frequencies are placed into unique buckets, with exact frequency counts. Each other class of items is placed into its own Count-Min Sketch data structure. For a schematic, see Figure 1. By separating the input items in this way, we can separately optimize the parameters of the CMS data structure for each class, allowing us to achieve a lower false positive rate given a fixed space budget for the $(\epsilon, k)$-frequent items problem. For example, for items with fairly high predicted frequencies, we may be able to use relatively small tables in the CMS data structure, without risking a large number of false positives, despite the high number of hash collisions.

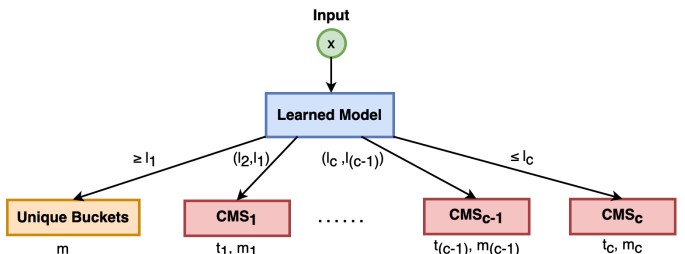

Figure 1: Partitioned Learned Count-Min Sketch (PL-CMS). Observe that items are partitioned based on scores returned by the learned model, with score thresholds denoted by $l_1, \ldots, l_c$. $t_i$, $m_i$ denote the number of tables and buckets for each Count-Min Sketch $CMS_i$. Our approach takes advantage of the learned model output space more extensively than the standard LCMS algorithm, which uses just a single score threshold.

PL-CMS is inspired by the work of Dai & Shrivastava (2020) and Vaidya et al. (2020) on learning augmented Bloom filters. Both papers enhance a simple learned Bloom filter by separating the output score space of a learned model into multiple regions according to different threshold values, and assigning the items in each region to a separate Bloom filter. They demonstrate that this approach leads to space savings over several Bloom filter variants, including standard, learning augmented with a single score threshold, and the sandwiched learned Bloom filter (Mitzenmacher, 2018).

**Parameter Optimization.** As with the partitioned learned Bloom filters of Dai & Shrivastava (2020) and Vaidya et al. (2020), a key challenge with PL-CMS is optimizing the multiple algorithm parameters, including the number of regions and the associated threshold values, and the space budget allocated to each CMS data structure. In the Bloom filter setting, Vaidya et al. (2020) demonstrate how to optimize these parameters via dynamic programming, given a fixed number of regions.

In the case of PL-CMS, parameter optimization is more difficult. Nevertheless, we show that given a fixed set of thresholds that partition the learned model output space, we can efficiently optimize the space allocations for each region's CMS data structure, as well as the parameters of these data structures, in order to minimize the false positive rate for frequent items detection. Specifically, we show how to optimize the number and size of the tables for each CMS data structure, assuming a fixed space allocation. We then apply an adjusted upper-bound formula for the theoretical false positive rate of CMS to optimize the space allocations themselves via a convex program.

**Empirical Performance.** We evaluate our algorithm on four datasets: *Macbeth*, *Bible*, Beijing PM2.5 concentration, and AOL search queries. We show that PL-CMS obtains a significantly lower false positive rate for frequent item detection, as compared to standard CMS and LCMS on datasets with heavy-tailed (Zipfian) frequency distributions. This is a commonly assumed input distribution in prior work (Hsu et al., 2019; Aamand et al., 2019; Zhang et al., 2020; Du et al., 2021). On datasets with very skewed, non-heavy-tailed frequency distributions, PL-CMS does not offer significant advantages, as there is not much to gain from further partitioning of the learned model ouput space. However, we demonstrate that PL-CMS still matches the performance of LCMS in these cases.

We remark that, while prior work focuses on various error metrics, such as the mean-squared frequency estimation error over all items in the stream (Zhang et al., 2020) or the average absolute estimation error (Hsu et al., 2019), we focus specifically on FPR for frequent item identification. In many applications, performance on this task is the key goal. An interesting direction for future research would be to develop algorithms that take advantage of the full learned model output space in order to improve performance on other frequency estimation error metrics.

## 2 PARTITIONED LEARNED COUNT-MIN SKETCH

In this section, we formally describe the design of the Partitioned Learned Count-Min Sketch (Section 2.1), and derive a theoretical upper bound for its expected false positive rate (Section 2.2). We will later use this upper bound in Section 3 to optimize the parameters of the algorithm in order to achieve a minimal false positive rate given a fixed space budget.

## 2.1 DESIGN

As discussed, the PL-CMS data structure assumes access to a learned model that is trained to predict frequencies of items in the input stream. We also assume a fixed set of score thresholds $l_1 > l_2 > \ldots > l_c$, which partition the learned model output space into $c + 1$ regions: all inputs with score $\geq l_1$, all inputs with scores in $[l_{i+1}, l_i)$ for $1 \leq i \leq c - 1$, and all inputs with score $< l_c$. Except for the first region with scores $\geq l_1$, each score region, which we number from 1 to $c$, is associated with a CMS data structure, denoted $CMS_i$. This data structure uses $t_i$ tables (associated with $t_i$ 2-universal hash functions), each with $m_i$ buckets.

When an item arrives in the stream, it is passed to the learned model, which determines which region it falls in. If its score is $\geq l_1$, then its frequency is tabulated in a 'unique bucket'. That is, when the item first arrives, a counter is allocated for it. On future arrivals, this counter is incremented in order to record the exact frequency of the item. Alternatively, if the item falls in score region $i \in \{1, \ldots, c\}$, it is inserted into $CMS_i$.

Assume that our algorithm is allocated a space budget of $S$ counters. Let $m$ denote the number of counters allocated to unique buckets, i.e., the number of unique items in the stream with predicted score $\geq l_1$. We will always assume that $l_1$ is set large enough such that $m < S$.

Let $S' = S - m$ denote the space remaining to allocate to our $c$ different CMS data structures. For simplicity, we will denote the space allocated to $CMS_i$ as $r_i \cdot S'$ where $r_i \in [0, 1]$ and $\sum_{i=1}^{c} r_i = 1$.

## 2.2 FALSE POSITIVE RATE UPPER BOUND

Determining the fraction of space $r_i$ allocated to each CMS data structure within PL-CMS, along with the parameters $m_i$ and $t_i$ of these data structures is crucial to the performance of the algorithm. Our aim is to optimize these parameters to obtain a low false positive rate for the $(\epsilon, k)$-frequent items identification problem. To do so, we first derive a closed form upper bound for the false positive rate of PL-CMS given a fixed space budget $S'$. This upper bound depends on the parameters $r_i, m_i$, and $t_i$ for $i \in \{1, \ldots c\}$. It also depends on several statistics of the input dataset. In particular:

1. The fraction of infrequent elements assigned to each region (i.e., the number of unique items assigned to that region with frequency $< (1 - \epsilon)\frac{n}{k}$ divided by the total number of unique items with frequency $< (1 - \epsilon)\frac{n}{k}$), which we denote by $F_i$

2. The total frequency of all items assigned to the region, which we denote by $E_i$.

As we will see in Section 3, exactly optimizing the parameters of PL-CMS requires knowledge of these statistics. However, in our empirical evaluation, we show that these quantities can be effectively approximated from historical or other training data, allowing for approximate optimization.

### 2.2.1 PER ITEM FALSE POSITIVE PROBABILITY

Consider an infrequent item in the data stream $x$ with true frequency $f_x < (1 - \epsilon) \cdot n/k$, which is placed in $CMS_i$ based on its learned model output score. We start by calculating an upper bound on the probability that $x$ is a false positive. Let $\hat{f}_{x,i,j}$ be the estimated frequency of $x$ in table $j$ of $CMS_i$. Let $h_{i,j}(\cdot)$ be the random hash function employed by this table, which maps input items to indices $\{1, \ldots, m_i\}$. As is standard in the literature (Cormode & Muthukrishnan, 2005), we assume that $h_{i,j}$ is 2-universal – i.e., for any $x \neq y$, $\Pr[h_{i,j}(y) = h_{i,j}(x)] \leq 1/m_i$. We have:

$$\mathbb{E}\left[\hat{f}_{x,i,j} - f_x\right] = \mathbb{E}\left[f_x + \left(\sum_{\substack{y \neq x \text{ s.t.} \\ h_{i,j}(y) = h_{i,j}(x)}} f_y\right) - f_x\right] \leq \frac{1}{m_i} \cdot E_i, \tag{1}$$

where $E_i$ denotes the sum of frequencies of items whose predictions place them in the region $i$. Applying Markov's inequality, we then have:

$$\Pr\left[\hat{f}_{x,i,j} - f_x \geq \epsilon \cdot \frac{n}{k}\right] \leq \frac{\frac{E_i}{m_i}}{\frac{\epsilon n}{k}} \leq \frac{k}{\epsilon \cdot m_i \cdot n} \cdot E_i. \tag{2}$$

Since we take a minimum counter over $t$ tables to obtain an element's frequency estimate in $CMS_i$, which we denote by $\hat{f}_{x,i} = \min_{j=1}^{t_i} \hat{f}_{x,i,j}$, we can upper bound the probability of having error at least $\frac{\epsilon n}{k}$ in this estimate by upper bounding the probability that all individual independent counters have large estimation error. That is:

$$\Pr\left[\hat{f}_{x,i} - f_x \geq \epsilon \cdot \frac{n}{k}\right] = \prod_{j \in [t_i]} \Pr[\hat{f}_{x,i,j} - f_x \geq \epsilon \cdot \frac{n}{k}] \leq \left(\frac{k}{\epsilon \cdot m_i \cdot n} \cdot E_i\right)^{t_i}, \qquad (3)$$

where we substitute the inequality from equation 2 to obtain the final bound.

Observe that since $x$ has frequency $< (1-\epsilon)\frac{n}{k}$, it can only be a false positive if $\hat{f}_{x,i} \geq \frac{n}{k} > f_x + \epsilon \cdot \frac{n}{k}$. Thus, equation 3 upper bounds the false positive probability for $x$ when assigned to $CMS_i$.

### 2.2.2 Expected Total False Positive Rate Upper Bound

We next compute an upper bound on the expected total false positive rate of PL-CMS by summing the false positive probabilities over all non-frequent elements and dividing the result by the number of unique infrequent elements $M$ in the data stream. For $i \in [c]$, let $B_i$ denote the set of infrequent items with frequency $< (1 - \epsilon)\frac{n}{k}$ assigned to $CMS_i$. Applying equation 3, we obtain:

$$FPR \leq \frac{1}{M} \cdot \sum_{i=1}^{c} \sum_{x \in B_i} \left(\frac{k}{\epsilon \cdot m_i \cdot n} \cdot E_i\right)^{t_i} \leq \sum_{i=1}^{c} F_i \cdot \left(\frac{k}{\epsilon \cdot m_i \cdot n} \cdot E_i\right)^{t_i}, \qquad (4)$$

where we recall that $F_i$ denotes the fraction of infrequent items falling into bucket $i$.

## 3 Parameter Optimization

We next show how to optimize the parameters of PL-CMS using the FPR upper bound derived in Section 2.2.2. Recall that, throughout, we assume a fixed set of thresholds defining $c$ regions. These thresholds in turn fix values for the statistics $E_i$ and $F_i$, which we assume knowledge of throughout. In Section 4 we demonstrate that in practice, sufficiently accurate approximations to these statistics can be obtained from training data, allowing effective optimization of PL-CMS.

We first optimize the parameters of each CMS while keeping space allocations for each region fixed (Sec. 3.1). We then show how to optimally allocate the space budget across regions (Sec. 3.2.

### 3.1 Optimizing CMS Parameters for Each Region

For each Count-Min Sketch $CMS_i$, given a fixed space allocation, we first show to to find the optimal number tables $t_i$ and buckets $m_i$ that minimize the bound on the total expected false positive rate given in equation 4. I.e., that minimize $F_i \cdot (\frac{k}{\epsilon m_i n} \cdot E_i)^{t_i}$. Suppose we allocate $s_i = r_i \cdot S'$ units of space to $CMS_i$. Then we can write $s_i = m_i \cdot t_i$. We substitute $m_i = \frac{s_i}{t_i}$ and take the derivative of our probability upper bound with respect to $t_i$, finding a root at $t_i = \frac{\epsilon s_i n}{e E_i k}$. Thus, we set:

$$t_i^* = \frac{\epsilon s_i n}{e E_i k} \qquad \text{and} \qquad m_i^* = \frac{s_i}{t_i} = \frac{e E_i k}{\epsilon n}. \qquad (5)$$

In the Appendix A.1, we conduct a series of experiments to verify that our theoretical optimization of $m_i$ and $t_i$ works well in practice.

### 3.2 Optimizing space allocations

We next show how the optimize the space allocations for each CMS data structure within PL-CMS, assuming a fixed total space budget $S'$. By plugging in the optimal $t_i^*$ and $m_i^*$ values from equation 5 into equation 4, we obtain the following bound for the expected total FPR of PL-CMS:

$$FPR \leq \sum_{i=1}^{c} F_i \cdot \left(\frac{1}{e}\right)^{\frac{\epsilon s_i n}{e E_i k}} = \sum_{i=1}^{c} F_i \cdot \left(\frac{1}{e}\right)^{\frac{\epsilon r_i S' n}{e E_i k}}. \qquad (6)$$

Observe that the only parameters that the above bound depends on are $r_1, \ldots, r_c$ – the fraction of space allocated to each $CMS_i$. Thus, it only remains to optimize these fractions. Our problem formulation becomes:

$$\min_{r_1, \ldots, r_c} \sum_{i=1}^{c} F_i \cdot \left(\frac{1}{e}\right)^{\frac{\epsilon r_i S' n}{e E_i k}} \text{ subject to: } r_i \geq 0, \text{ for } i = 1, \ldots, c \text{ and } \sum_{i}^{c} r_i = 1.$$

One can observe that the above function is convex in $r_1, \ldots, r_c$ and that the constraints are convex. In particular, we can formulate the problem as a conic exponential optimization instance, and solve it using e.g., the Mosek convex solver (ApS, 2022).

In practice, we find the bound of equation 6 is somewhat loose. Specifically, optimizing the space allocation using this theoretical upper bound does not necessarily optimize the FPR in practice. To fix this issue, we adjust the FPR bound by introducing a small constant $p > 1$ as follows:

$$\text{FPR} \approx \sum_{i=1}^{c} F_i \cdot \left(\frac{1}{e}\right)^{\frac{p \epsilon r_i S' n}{e E_i k}}. \tag{7}$$

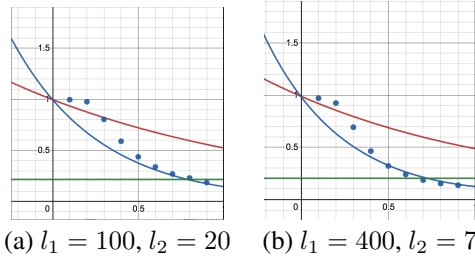

(a) $l_1 = 100, l_2 = 20$    (b) $l_1 = 400, l_2 = 7$

Figure 2: Comparison of the actual (equation 6) vs. adjusted (equation 7) FPR bound with the purpose of optimizing space allocation values for 2-CMS using the *Macbeth* dataset with total space $= 750$ (excluding the space reserved for unique buckets) where $l_1, l_2$ denote the first and second threshold respectively. The x-axis represents the fraction of space $r_2$ reserved for $CMS_2$. Note, $r_1$ is determined by $r_2$. The red and blue lines represent the theoretical and adjusted (with $p = 3$) FPR bounds, respectively. The green line represents the lowest FPR obtained for a single learned Count-Min Sketch in practice. The blue data points are FPR values obtained in practice by varying the value of $r_2$.

In Figure 2, we observe that using the adjusted FPR bound of equation 7 with a value of $p = 3$ to optimize the space allocations best approximates the optimal FPR values obtained in practice for the *Macbeth* dataset. We will later see that this constant works well across different datasets. Future research studying this adjustment from a theoretical standpoint could yield valuable insight.

## 4    EXPERIMENTS

We next evaluate the performance of our PL-CMS algorithm on several datasets, comparing the false positive rate in the task of identifying $(\epsilon, k)$-frequent items to standard CMS and the learned CMS approach of Hsu et al. (2019), which we denote as LCMS. Note, LCMS is equivalent to PL-CMS with a single Count-Min Sketch data structure. We show that in many cases, PL-CMS can obtain significantly lower false positive rates with the same space usage, as compared to existing approaches.

### 4.1    METHODOLOGY

We focus on evaluating PL-CMS with two or three learned model output score thresholds, and in turn two or three CMS data structures. We call these variants 2-CMS and 3-CMS, respectively. We find that using a larger number of score thresholds typically does not significantly improve performance.

To establish a baseline, we first test the partitioned variants assuming exact knowledge of the total frequency $E_i$ and fraction of infrequent items $F_i$ for each bucket, along with the exact number of

unique buckets $m$. This allows us to exactly optimize the parameters of PL-CMS using the approach of Section 3. We denote the resulting idealized algorithms as 2-CMS-A and 3-CMS-A.

In Section 4.4, we show how to estimate $E_i$, $F_i$, and $m$ effectively from training data. We then perform parameter optimization using these estimated values. We refer to the resulting algorithms as 2-CMS-E and 3-CMS-E, and demonstrate that they perform similarly to the idealized variants.

We compare all algorithms (PL-CMS, LCMS, and CMS) using various fixed space budgets of $s$ counters. This includes counters for both the CMS data structures and the unique buckets for the learned variants. We ignore the space of the learned model itself, which is lower order in our settings.

For each dataset, we pick two different values of $\frac{n}{k}$, the cut-off for defining a 'frequent item'. We set these cutoffs to the 95-th and 99-th percentiles in the dataset's frequency distribution. We test the algorithms under a wide range of space budgets, which yield a wide range of false positive rates. For PL-CMS and LCMS, we perform a grid search to pick learned model output threshold values that work well in practice. For LCMS we compare three values for the first threshold $l_0$ by scaling three relatively high frequencies in the training dataset's distribution which would likely yield good results for identifying $n'/k$ frequent items with $n'$ denoting the length of the training set. For 2-CMS, we use the same set of $l_0$ values as LCMS and add a second cutoff value $l_1$ by targeting the middle-frequency values of the training dataset. In the case of 3-CMS, we maintain the same two thresholds as 2-CMS and pick a relatively small frequency for $l_2$. For each variant, we report the false positive rates in the task of identifying $(\epsilon, k)$-frequent items returned by threshold settings which return the lowest average FPR across different space settings. For 2-CMS-E and 3-CMS-E we use the same sets of thresholds as 2-CMS-A and 3-CMS-A, respectively..

## 4.2 DATASETS

We conduct our experiments on several datasets. The first two, *Macbeth* and *Bible* are text datasets with Zipfian-like heavy-tailed frequency distributions, obtained from the NLTK Python library (Bird et al., 2009).*Macbeth* contains $3,446$ unique tokens with $18,292$ total occurrences. *Bible* contains $12,567$ unique tokens, with $792,026$ total occurrences. For our training data for the learned oracle and estimation of $E_i$ and $F_i$, we employ several other text datasets from the NLTK library. In particular, Jane Austen's *Emma*, a collection of poems by William Blake, and children's stories by Sarah Cone Bryant, totaling $215,783$ words and $9,502$ unique word tokens.

Our third data set is Beijing PM2.5 (Chen, 2017), which was previously used by Zhang et al. (2020) to evaluate learning-augmented CMS. This dataset covers a 4-year period and includes PM2.5 concentration values (treated as frequencies) for each day, along with other features like temperature, pressure, and wind speed. After preprocessing, the dataset comprises $41,757$ observations (i.e., unique items), with a cumulative PM2.5 concentration (i.e., total frequency) of $4,117,792$.

Our final dataset is an AOL search query dataset, as employed by Hsu et al. (2019), which comprises search queries collected over 90 days, accompanied by their respective frequencies (Huang, 2006). Due to space constraints, we relegate further details and results on this dataset to Appendix A.3.2.

## 4.3 LEARNED MODELS

Our evaluation focuses on the relative performance of our PL-CMS algorithm as compared to the existing LCMS approach of Hsu et al. (2019). As such, we employ relatively simple learned frequency estimation models within these algorithms.

For the Zipfian datasets (*Macbeth* and *Bible*), we simply predict the frequencies of words in the literary text datasets by utilizing Zipf's Law and assuming that we know the length of the datasets $n$. In particular, we utilize a training dataset to predict the word ranking of an incoming item and use it to obtain a frequency estimate resulting from Zipf's Law. We describe our predictor in more detail in the Appendix A.2. The Zipfian oracle obtains a mean absolute prediction error (MAE) of $3.79$ and $44.77$ on the *Macbeth* and *Bible* datasets, respectively.

For the PM2.5 dataset, we choose several key features including date, dew point, temperature, pressure, combined wind direction, and cumulative wind speed to input into our learned model. We divide the dataset into $20\%$ training and $80\%$ test sets and use *Scikit-learn* (Pedregosa et al., 2011) to train a straightforward neural network with three hidden layers of sizes 100,100 and 20 to make

predictions of the PM2.5 values (i.e., the frequencies) using the other parameters. We use the *Adam* optimizer (Kingma & Ba, 2017) and ReLU as the activation function. The resulting model achieves a regression score of 0.488 and an MAE of 43.50.

## 4.4 ESTIMATING $E_i$, $F_i$, AND $m$

Recall, that the exact algorithms 2-CMS-A and 3-CMS-A require that we know the proportion of infrequent items $F_i$, and the total frequency of items $E_i$ within each region $i$ as well as the value of $m$ for the process of parameter optimization. To estimate the first two quantities we make an assumption that we have prior knowledge of the total stream length, denoted as $n$. In practice, one could approximate $n$ using past data.

Here we focus on the estimation process for Zipfian datasets (*Macbeth* and *Bible*) – a similar approach can be applied to the remaining datasets. Unlike the text corpora, the latter datasets do not require approximating the number of unique items $u$ of the testing dataset in order to estimate $m$.

Specifically, for the Zipfian datasets, we first use Heap's law (Herdan, 1960) which approximates the true number of unique words by $D(n) \approx K \cdot n^{\beta}$, where $n$ represents the dataset length, and $K$ and $\beta$ are free parameters. Typically, $\beta$ falls in $[0.4, 0.6]$. We use the training dataset to fit these two parameters and then apply the parameters to estimate the number of unique words in the test dataset, given the total number of items $n$.

We next approximate $E_i$ and $F_i$ for each region of the testing dataset. To do so, we scan the training dataset and record $E'_i$ and $F'_i$ values using adjusted values for the score thresholds as well as the heavy-hitter cut-off $(1 - \epsilon) \cdot n/k$ (which is relevant to computing $F_i$). In particular, we scale all of these values by a factor of $n'/n$, where $n'$ is the length of the training dataset, so that they represent the same *relative frequency* in the testing and training datasets. We also record a unique bucket count $m'$ for the training dataset, using the same adjusted thresholds.

To obtain estimates for $E_i$, $F_i$, and the number of unique buckets $m$ for the testing dataset, we then scale back our estimates: $E_i \approx E'_i/n' \cdot n$ and $m \approx m'/u' \cdot D(n)$, where $u'$ denotes the true number of unique words in the training dataset. Since $F'_i$ is a fraction, we expect that $F'_i$ roughly approximates $F_i$, so $F_i \approx F'_i$. Note, for the PM2.5 and AOL datasets, we can simply use $m \approx m'/n' \cdot n$ as an estimate of $m$ since the ratio of unique items in the training and testing datasets is roughly proportional to their respective lengths (which is not true for the Zipfian datasets). It is also worth mentioning that, in case these procedures underestimate the true count $m$ of the most heavy items, we store the remaining elements in $CMS_1$.

## 4.5 RESULTS

***Macbeth.*** For this dataset, we set $\frac{n}{k} = 16$ and $\frac{n}{k} = 80$. For $\frac{n}{k} = 16$, we use the following set of thresholds for 2-CMS-A and 3-CMS-A, respectively: $(l_1 = 200, l_2 = 100)$ and $(l_1 = 300, l_2 = 200, l_3 = 100)$. We demonstrate that both actual and estimated 2-CMS outperform the standard and learned CMS in terms of the FPR by more than $5\%$ on average (Figure 3). In addition, both 3-CMS variants achieve a roughly $10\%$ lower false positive rate as compared to CMS and LCMS. We note that 3-CMS assigns most of its space to $CMS_3$ and sets both $m_1$ and $m_2$ to 1. Although it accrues a lot of error in the first two regions, the true fraction of infrequent items in $CMS_i$ and $CMS_2$ is relatively low and the total frequencies falling into these regions is relatively high. Thus, by filtering the heavy items from $CMS_3$ we are able to obtain a lower overall FPR. We include results for setting $\frac{n}{k} = 80$ in the Appendix A.3.1, in which we show that the 3-CMS variants outperform the rest by $10\%$ on average.

***Bible.*** For our experiments, we set $\frac{n}{k} = 118$ and $\frac{n}{k} = 879$. For the first setting we set the thresholds for 2-CMS-A and 3-CMS-A as: $(l_1 = 3000, l_2 = 250)$ and $(l_1 = 2000, l_2 = 300, l_3 = 250)$. Figure 3 shows that for $\frac{n}{k} = 118$, all partitioned heuristics outperform CMS and LCMS by about $10\%$ in terms of the false positive rate. When we set $\frac{n}{k} = 879$, given thresholds $(l_1 = 3000, l_2 = 250)$ and $(l_1 = 2000, l_2 = 400, l_3 = 250)$ for 2-CMS-A and 3-CMS-A, respectively, PL-CMS gives a substantial advantage in FPR in comparison to the baseline algorithms – from a $10\%$ to over $20\%$ reduction in FPR as we decrease the total space. We observe that in both these settings 3-CMS performs similarly to 2-CMS. This is due to the dataset being larger and hence having a more skewed distribution, where adding another partition to PL-CMS to separate middle-frequent elements does

not offer any advantages in identifying heavy hitters with much higher relative frequencies. In particular, both 2-CMS and 3-CMS allocate almost all of the space budget for the items in the last region with predicted frequency less than $l_c$ since it exhibits the largest predicted fraction of infrequent items. At the same time, 3-CMS' second Count-Min Sketch $CMS_2$ does not contain a sufficiently large amount of frequencies in order to lower the FPR in $CMS_3$, yielding a similar performance to 2-CMS.

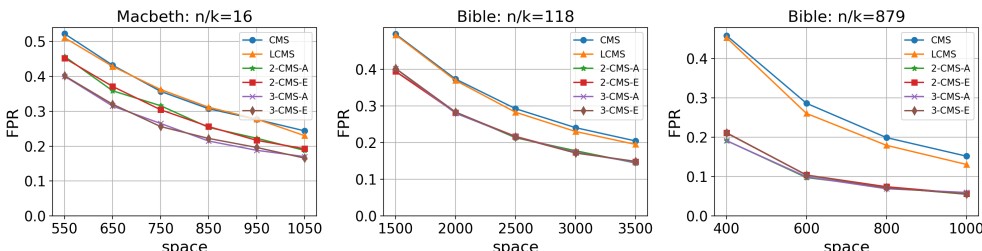

Figure 3: False positive rate vs. space for *Macbeth* and *Bible* datasets with different $\frac{n}{k}$ values.

**PM2.5.** In Figure 4 we notice that PL-CMS substantially outperforms the baseline algorithms for smaller space allocations and lower values of $\frac{n}{k}$. We observe that for $\frac{n}{k} = 284$, with thresholds ($l_1 = 300$, $l_2 = 200$, $l_3 = 100$), 3-CMS obtains a 22% lower FPR using 2000 units of space. Since CMS exhibits a large amount of collisions in a limited space setting, separating the elements into different Count-Min Sketch data structures yields a lower FPR. For larger space allocations, 3-CMS yields a similar amount of infrequent items in $CMS_2$ and $CMS_3$ with relatively large amounts of total frequencies in each, in which case it is cheaper to store all the elements in a single CMS.

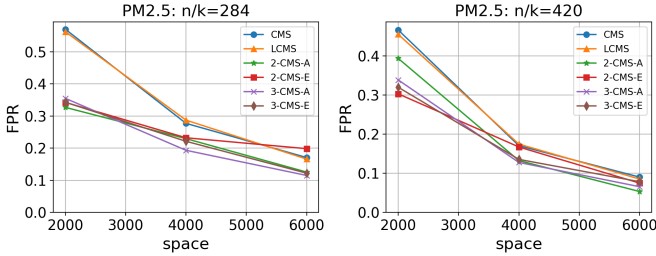

Figure 4: False positive rate vs. space for the PM2.5 dataset with different $\frac{n}{k}$ values.

## 5 CONCLUSION

We have introduced the Partitioned Learned Count-Min Sketch (PL-CMS) variant, which leverages the comprehensive output space score of a learned oracle to enhance the performance of a standard CMS. We have proposed a theoretically motivated heuristic, which given a fixed set of thresholds defining the partitions of PL-CMS, optimizes space allocation for each backup data structure and its parameters. Our empirical results demonstrate that it significantly outperforms the baseline algorithms in the task of identifying $(\epsilon, k)$-frequent items on datasets with a Zipfian distribution using very simple learned models.

Future research is needed to tighten the analysis for the upper bound on the false positive rate returned by PL-CMS as well as to find a way to optimize the threshold values with respect to the parameter $k$. Furthermore, it's important to note that our experiments with PL-CMS are limited to 3-CMS. Determining the optimal number of partitions is a crucial next step that can significantly enhance its performance. It is also worth mentioning that the approach of partitioning the learned model's output score space can be extended to any frequent item estimation algorithm, including Count-Sketch. Nevertheless, future study is needed to fine-tune the parameters of such a learned variant.

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

## A    APPENDIX

### A.1    OPTIMIZING CMS PARAMETERS FOR EACH REGION IN PRACTICE

In this section, we verify the formulas for setting $m_i$ and $t_i$ (equation 5) for each Count-Min Sketch $i$ from Section 3.1. To do that, we run a series of experiments using the LCMS algorithm of Hsu et al. (2019). LCMS is equivalent to PL-CMS with a single threshold at $l_1$ and thus a single Count-Min Sketch $CMS_1$ with parameters $t_1$ and $m_1$, that is allocated $s_1$ units of space. We conduct the experiments on the *Macbeth* dataset, which is a text corpus dataset of Shakespeare's work containing $n = 18,292$ words (Bird et al., 2009).

Using different settings for the $CMS_1$ space usage $s_1$ and the single learned model output threshold $l_1$, we compute $t_1^*$, the optimal $t_1$ value given by the formula in equation 5, using exact values of $F_1$ and $E_1$. We then range $t_1$ from 1 to 10 and calculate the FPR in the task of identifying $(\epsilon, k)$-frequent words from *Macbeth* with $n/k = 26$ and $\epsilon = 0.5$. We run the experiment with 10 repetitions and report the average FPR values for each $t_1$. In Table A.1, we observe that rounding up the $t_1^*$ values computed by optimizing the FPR upper bound for space allocations 750 and 2000 achieves the optimal false positive rate in practice. With 5000 units of space, we get near-optimal results – $t_1^*$ achieves a slightly larger FPR than the actual optimal $t_1$ value. These findings indicate that setting $t_i$ and $m_i$ using equation 5 generally works well in practice.

| **LCMS** | $s_1 = 750, l_1 = 100$ $t_1^*$=0.4868 | $s_1 = 2000, l_1 = 200$ $t_1^*$=1.2305 | $s_1 = 5000, l_1 = 400$ $t_1^*$=2.8760 |
|---|---|---|---|
| | **Avg FPR** | **Avg FPR** | **Avg FPR** |
| $t_1 = 1$ | **0.2172** | 0.0663 | 0.0259 |
| $t_1 = 2$ | 0.3085 | **0.0546** | 0.0175 |
| $t_1 = 3$ | 0.5421 | 0.0669 | 0.0150 |
| $t_1 = 4$ | 0.7816 | 0.0678 | 0.0143 |
| $t_1 = 5$ | 0.9289 | 0.1100 | **0.0138** |
| $t_1 = 6$ | 0.9852 | 0.1670 | 0.0143 |
| $t_1 = 7$ | 0.9997 | 0.2431 | 0.0139 |

Table 1: Comparison of $t_1$ values optimized using the FPR bound (equation 5), denoted as $t_1^*$, and $t_1$ values found in practice for *Macbeth* dataset using LCMS with space $s_1$ and threshold $l_1$. The bolded FPR values represent the lowest FPR values for a given space and cutoff setting. The boxed values represent the FPR values achieved by $t_1^*$ rounded up to the closest integer. Notice, the boxes coincide with the bolded values in the first two columns.

### A.2    ZIPF'S LAW AND THE LEARNED MODEL FOR THE ZIPFIAN DATASETS

We leverage Zipf's law to predict the frequencies of words in the *Macbeth* and *Bible* datasets. The law states that given a large sample of words, the frequency of any word is inversely proportional to its rank in the frequency table. Specifically, we have that for any element $x$ from a distribution, the fraction of times it occurs is given by:

$$f(x) = \frac{1}{r_x^s H_{N,s}}$$

where $r_x$ is the rank of $x$, $s$ is the value of the exponent characterizing the distribution and $N$ the number of elements in this distribution. The value $H_{N,s}$ represents the $N$-th generalized harmonic

number. As for human languages, word frequencies can be modeled well by a Zipfian distribution with $s = 1$. Thus, we predict the frequency of a word using the following formula:

$$F(x) = \frac{1}{H_{N,1} \cdot r_x} \cdot n,\tag{8}$$

where $n$ is the total length of the dataset and $N = 171,476$ is the approximate number of words in the English language given by the *Oxford English Dictionary*. Using the literary works chosen for our training dataset, we compute a word ranking by tracking the number of times each word appears. Then, we calculate frequency predictions for items in our testing dataset according to equation 8 with adjusted $n$ values. In Figure 5 we observe that the model roughly approximates the true frequencies of words in the two text corpora.

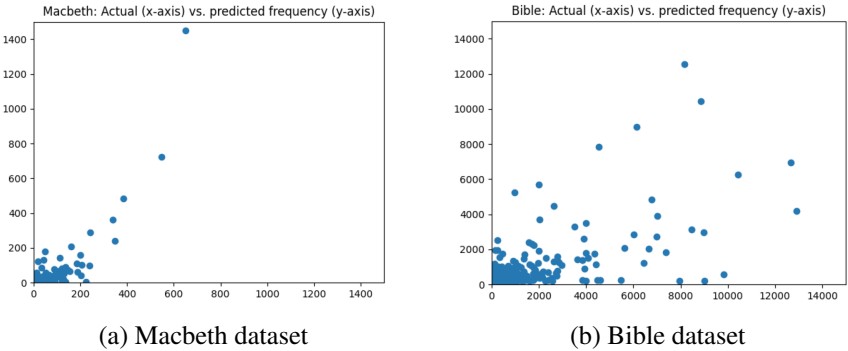

(a) Macbeth dataset           (b) Bible dataset

Figure 5: Predicted versus actual frequencies of words in the text corpora.

## A.3 ADDITIONAL RESULTS

In this section, we include additional results for the *Macbeth* dataset as well as another real-world dataset, AOL.

### A.3.1 *Macbeth*

For setting with $\frac{n}{k} = 80$, we use the following set of thresholds for 2-CMS and 3-CMS, respectively: $(l_1 = 400, l_2 = 80)$ and $(l_1 = 400, l_2 = 80, l_3 = 20)$. In Figure 6 we observe that, although 2-CMS-A and 2-CMS-E perform similarly to 1-CMS and LCMS, both 3-CMS variants outperform the former by over $10\%$ across different space settings.

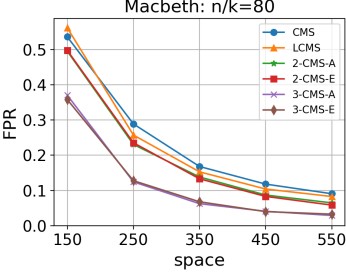

Figure 6: False positive rate vs. space for the *Macbeth* dataset with $\frac{n}{k} = 80$.

### A.3.2 AOL

**Dataset.** We include results for the AOL search query data, as employed by Hsu et al. (2019). This dataset comprises search queries collected over 90 days, accompanied by their respective lookup

frequencies (Huang, 2006). We utilize the data collected on the first day consisting of $204,431$ unique items with a combined frequency of $372,763$ to predict the frequencies of $185,296$ queries made on the second day, which collectively amounted to a total length of $339,199$.

**Learned Model.** Regarding our AOL model, we opted not to employ the model introduced by Hsu et al. (2019) due to its high accuracy, which resulted in PL-CMS not surpassing LCMS that correctly identifies all frequent items via an oracle. Instead, we adopted a different approach by leveraging the *fasttext* library for search query vectorization, along with the *Scikit-learn* toolkit. We trained a simple classifier using data from the initial day to predict query frequencies on the following day. The classifier uses a single hidden layer with 100 neurons, the ReLU activation function and the *Adam* optimizer, yielding a classification score of 0.69.

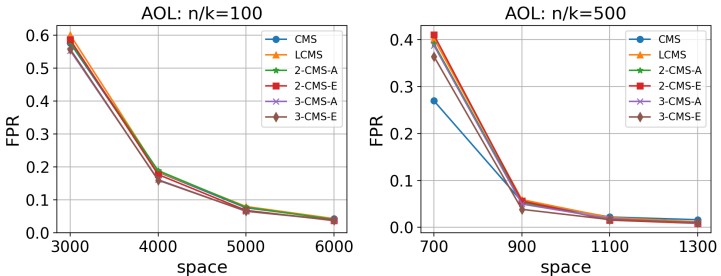

Figure 7: False positive rate vs. space for AOL dataset with different $\frac{n}{k}$ values.

**Results.** In Figure 7 we observe that PL-CMS does not outperform the baseline algorithms, but reports similar false positive rates as the standard CMS for larger space allocations. In the case of small space budgets, the standard CMS obtains a significantly lower FPR than PL-CMS in the task of identifying very frequent items with $n/k = 500$. This is due to the data set being very skewed since roughly $95\%$ of the queries appear only once in the stream and, as a result, it is easy to identify elements with high frequency using just the standard CMS variant.

### A.3.3 ROBUSTNESS

In this section, we study the empirical performance of PLCMS under random predictions. Specifically, fixing the number of unique buckets, we let the oracle assign the elements of the data stream uniformly into different regions. As previously, we use the training dataset to predict $E_i$ and $F_i$ values. In Figure 8, we observe that PLCMS matches the performance of CMS and LCMS except for 3-CMS on the Macbeth dataset with the $n/k = 16$ setting.

### A.3.4 COMPARISON TO COUNT-SKETCH

In this section, we provide a comparison of PL-CMS to PL-CS, where the latter uses the Count-Sketch algorithm instead of Count-Min Sketch. In all the experiments we use the same parameters for PL-CS as for PL-CMS using the actual $E_i$ and $F_i$ values, since we do not know how to optimize its parameters to minimize the FPR. We also apply the same best cutoff thresholds.

In Figure 9 we observe that the standard and learned CS of Hsu et al. (2019) (denoted as CS and LCS, respectively) perform similarly to basic CMS and LCMS. For the Macbeth dataset with $n/k = 16$, PL-CS with 3 Count-Sketches (denoted as 3CS-A) achieves a worse FPR than the baseline algorithms and 3-CMS-A. For Bible with $n/k = 879$, 2CS-A also underperforms and yields a much higher FPR than the baseline algorithms and 2CMS-A. It is also important to mention that using Count-Sketch induces a small false negative rate in the task of identifying the frequent elements which is not present for Count-Min Sketch variants. While we do not expect the optimal PL-CS to provide significant advantages as compared to PL-CMS, optimizing its parameters is an interesting future research direction.

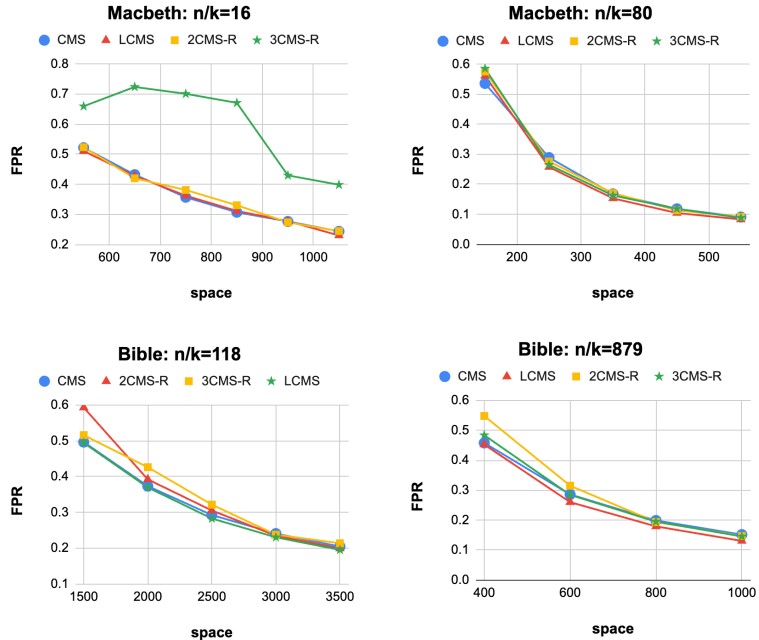

Figure 8: Comparison of the false positive rates obtained from 2CMS and 3CMS with random oracles denoted at 2CMS-R and 3CMS-R, respectively with the baseline CMS and LCMS algorithms.

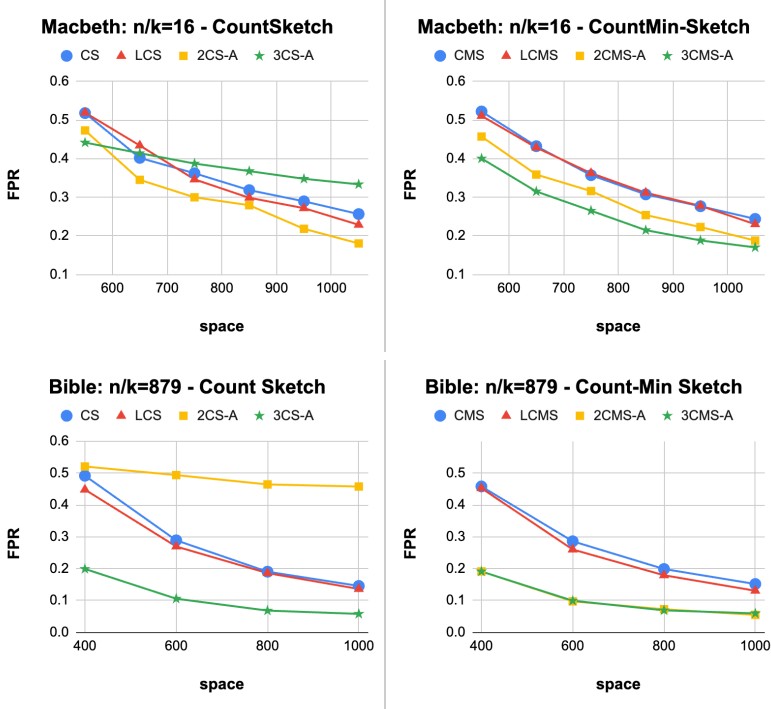

Figure 9: Comparison of the false positive rates obtained from Partitioned Learned Count-Sketch (PL-CS) and PL-CMS using the same parameters and threshold values as well as the exact $E_i$ and $F_i$ values. CS denotes standard Count-Sketch, LCS denotes learned Count-Sketch of Hsu et al. (2019), 2CS-A and 3CS-A denote PL-CS with two and three Count-Sketch data structures, respectively.

