# OpenReview forum: "Partitioned-Learned Count-Min Sketch"
_ICLR.cc/2024/Conference — Submitted to ICLR 2024_

### Official Review · Reviewer_C7mx · 2023-10-28

**Soundness:** 2 fair
**Presentation:** 3 good
**Contribution:** 2 fair
**Rating:** 5
**Confidence:** 4

**Summary:**

This paper aims to incorporate learned information to improve the performance for the count-min sketch (for the heavy-hitter analysis task in data streams). In particular, the algorithm receives a prediction of the frequency of input items, and it utilizes this info to build a separate data structure for items with similar (predicted) frequencies via a simple thresholding strategy. Indeed, if this partition of the data is good enough (which means items of similar frequency are put in the same parts), then it can achieve a better accuracy-space tradeoff compared with a generic count-min sketch. Similar ideas have been considered in recent papers such as Vaidya et al. (ICLR 2021) which studied bloom filters.

**Strengths:**

I think that the general idea of having separate count-min sketch for items of similar frequency is nice, and it is convincin that this could improve the performance (provided that the prediction is accurate). In addition, the experiment results seem to be promising. In particular, it looks like even with a relatively simple and weak prediction, the proposed algorithm can already achieve significant improvement over baselines on various data sets.

**Weaknesses:**

- The algorithm needs additional information than the estimated frequency of items, particularly the E_i F_i parameters are only artifacts of your algorithm instead of something natural to the heavy-hitter problem

- The robustness (i.e., what happens if the prediction is completely wrong) is not discussed/evaluated. Indeed, one major motivation for algorithms with predictions is to utilize ML predictions while still preserving the worst-case guarantee.

- It would also be better to have a measure of the prediction error, as well as relate the performance of your algorithm to that error measure to obtain a smooth tradeoff between robustness and consistency.

- I also see some technical issues, and please find the detailed comments in the "Questions" section.

**Questions:**

- It seems your E_i and F_i are dependent on the thresholds t_i’s. However, it seems you need to estimate E_i and F_i, and then use them to find the t_i’s. This does not make sense to me.

- Page 4, “which region is falls in” -> “which region it falls in”

- Page 4, item 1 in Sec 2.2 has unpaired parenthesis

- Page 6, third paragraph. Can you give more intuition on why the theoretical upper bound does not work well? Is there any rationale for introducing a p in the exponent? This way of adding a new parameter seems quite random to me.

- In page 6 you mentioned that using larger number of score thresholds typically does not improve the performance — why is this? Also, this somewhat contradicts the claim in Section 5, where you mention that “determining the optimal number of partitions is a crucial next step that can significantly enhance its performance”.

- Page 7, why the space of the learned model is of the lower order so that it can be ignored?

- Only the arXiv version of Vaidya et al., ICLR 2021 paper is cited — consider citing the conference version. Please also check this for other references.

- Several references do not have their venues listed, for isntance Chabchoub et al., 2009 and Dolera et al., 2022.

---

> ### Author Response · Authors · 2023-11-18
> **Rebuttal by Authors**
>
> Thank you for your constructive feedback and careful reading of our paper. We address the most important questions and concerns below.
> > The robustness is not discussed/evaluated. Indeed, one major motivation for algorithms with predictions is to utilize ML predictions while still preserving the worst-case guarantee.
>
> Given fixed thresholds, even when the predictions are completely inaccurate, PL-CMS optimizes the space allocations of each CMS accordingly, which means that PLCMS is naturally robust. Specifically, we use the predictions to estimate the quantities $E_i$ and $F_i$ as well as the number of unique buckets from the training dataset. If, for example, we incorrectly place a large number of infrequent items in the first region, $F_1$ will be large and we will assign $\text{CMS}_1$ a lot of space to lower its FPR.
>
> As an example, consider the case when our ML model is just a random oracle that uniformly assigns elements of the stream into different regions of PL-CMS, without any correlation to their actual frequencies. Further, assume that the number of unique buckets is small (which is typically true in practice). In this setting, one can show that the expected error of any item under PL-CMS is roughly equal to its expected error under the standard Count-Min sketch -- that is, when the learned model degenerates to random guesses, PL-CMS still matches the performance of the unlearned algorithm, indicating a high level of robustness. Since a random oracle will allot every region a roughly equal fraction of infrequent items $F_i$ and a roughly equal total frequency of items $E_i$ and, as a result, each CMS will be assigned the roughly same amount of buckets $m_i$, we will have $\frac{E_i}{m_i}\approx \frac{n/c}{m/c} = \frac{n}{m}$, where $c$ is the number of regions. This is the same ratio (and in turn yields the same false positive rate) as if we assigned all items to a single CMS with $m$ buckets. Note that a similar argument would hold if the model assigned items to different regions with different probabilities. In this case, letting $p_i$ be the probability of an item being assigned to region $i$, we would have $m_i \approx m \cdot p_i$.
>
> To verify the above argument, we have included additional experiments in section A.3.3 of the Appendix, where we show that given a learned model that outputs random predictions, PL-CMS performs similarly to standard CMS and LCMS on the Zipfian datasets except for 3-CMS on the Macbeth dataset with $n/k=16$ which we are currently investigating.
> We would also like to point out that while it is important to consider the robustness of algorithms with predictions, none of the prior works in the area (LCMS of Hsu et al. and Partionioned Learned Bloom Filters of Vaidya et al.) provide such an analysis.
> We agree that establishing a measure of prediction error is a crucial next step to optimizing the threshold values of PL-CMS and studying its consistency.
>
> > It seems your $E_i$ and $F_i$ are dependent on the thresholds $t_i$’s. However, it seems you need to estimate $E_i$ and $F_i$, and then use them to find the $t_i$’s. This does not make sense to me.
>
> The threshold values do not depend on $E_i$ and $F_i$. We first perform a grid search to find the threshold values and then estimate $E_i$ and $F_i$ given the chosen thresholds.
>
> > Page 6, third paragraph. Can you give more intuition on why the theoretical upper bound does not work well? Is there any rationale for introducing a $p$ in the exponent? This way of adding a new parameter seems quite random to me.
>
> The theoretical bound on the false positive rate is loose due to applying Markov's inequality in the analysis. The term inside the parentheses found in Equation 6 is the failure probability for any repetition which is set to $1/e$ by picking $m_i$ appropriately. We then raise it to the number of repetitions $t_i$. Thus, adding a constant $p$ is equivalent to estimating the per repetition failure probability as $1/e^p$ instead of $1/e$ which makes it closer to reality.
>
> > In page 6 you mentioned that using larger number of score thresholds typically does not improve the performance — why is this? Also, this somewhat contradicts the claim in Section 5.
>
> This is a great question. We agree that optimizing the thresholds themselves rather than the number of regions is more important to improving our algorithm's performance. Finding a way to optimize the number of regions could potentially allow us to better understand why using a small number of regions often works well in practice.
>
> > Page 7, why the space of the learned model is of the lower order so that it can be ignored?
>
> Thank you for pointing this out. We follow the approach of Hsu et al. in which we focus on the space complexity of the data structure instead of its learned model since the space occupied by the oracle can be amortized over time. We will include a clarification in the paper.

---

### Official Review · Reviewer_4TXw · 2023-11-01

**Soundness:** 3 good
**Presentation:** 3 good
**Contribution:** 2 fair
**Rating:** 6
**Confidence:** 3

**Summary:**

This paper studies how to partition the stream into multiple region and process each local region with a Count-Min Sketch. Heavy hitters and frequent items are important tasks in streaming setting and have many applications. The authors provide both theoretical analysis and experimental studies to showcase the proposed algorithm outperforms the baseline in many cases.

**Strengths:**

The main idea of this paper is sound and intuitive. In fact many other sketching algorithm (not learned) leverage the same idea. Some recent works in the space are AugmentedSketch, ElasticSketch, and Panakos.

The theory analysis is well written and friendly to readers. The analysis looks correct to me.

The parameter optimization method is interesting and may lead to broader impact.

**Weaknesses:**

AOL dataset is subject to controversy. I would recommend the author to remove experimental results about AOL. (https://en.wikipedia.org/wiki/AOL_search_log_release)

Author may want to clarify the assumption on the stream and compare with some other popular summaries in the experiments to indicate the benefits of learning. If the stream is in insertion-only or in bounded-deletion model, then author should compare with the SpaceSaving algorithm. (see https://arxiv.org/pdf/2309.12623.pdf and https://arxiv.org/abs/1803.08777). If the stream is in turnstile model, then the author should include comparison with Count Sketch (Charikar, Moses, Kevin Chen, and Martin Farach-Colton. "Finding frequent items in data streams.").

I might have missed it. How is the score $l_i$ decided and is it learned in training? For instance, in Macbeth, the proposed algorithms use threshold 200|100, and 300|200|100.

**Questions:**

See weakness

---

> ### Author Response · Authors · 2023-11-18
> **Rebuttal by Authors**
>
> We thank you for your positive review and helpful comments. We address the main questions and concerns below:
>
> > AOL dataset is subject to controversy. I would recommend the author to remove experimental results about AOL.
>
> Thank you for pointing this out -- we were not aware of this controversy. We initially included the results for AOL to be able to compare our algorithm with LCMS of Hsu et al. which reports results for the same dataset. We will remove it and look for a different search log dataset to replace it with.
>
> > Author may want to clarify the assumption on the stream and compare with some other popular summaries in the experiments to indicate the benefits of learning. If the stream is in insertion-only or in bounded-deletion model, then author should compare with the SpaceSaving algorithm. If the stream is in turnstile model, then the author should include comparison with Count Sketch.
>
> In all the applications we consider, the stream is insertion only. However, our algorithm in principle can work in turnstile streams as well. Moreover, PL-CMS can be used with any base frequency estimation sketch, as long as we can derive a closed-form formula for the FPR that we can optimize to determine space allocations across the regions. For the Count-Min sketch, this formula leads to a convex program that we can efficiently solve. We have attempted to extend our approach to Count sketch but were not able to find a formula that we could efficiently optimize. Nonetheless, we agree that including an empirical comparison to other frequency estimation algorithms and their partitioned learned variants would be highly interesting.
>
> > I might have missed it. How is the score decided and is it learned in training? For instance, in Macbeth, the proposed algorithms use threshold 200|100, and 300|200|100
>
> This is a great question. We choose the thresholds by performing a grid search on the training dataset and picking thresholds that yield the lowest average FPR in identifying $(\epsilon,k)$-frequent items in a wide range of space allocations. Assuming the length of the testing dataset $n$, we scale the found thresholds by $n/n_{tr}$ where $n_{tr}$ is the length of the training dataset. Specifically, to find the first threshold $l_1$, we incrementally iterate through a wide range of cutoff values and report the one that yields the lowest FPR for LCMS on the training dataset. Next, we fix $l_1$ and perform another grid search to find the next cutoff $l_2< l_1$ which yields the lowest FPR for 2-CMS. Similarly, to find $l_3$ (where $l_3 < l_2$), we fix $l_1$ and $l_2$. We let $l_3$ be the cutoff value which yields the lowest FPR for 3-CMS using $l_1$,$l_2$ and $l_3$. In the final step, we rescale the thresholds by $n/n_{tr}$. We will include a more detailed description of the process in our paper.

---

> ### Author Response · Authors · 2023-11-21
> **Rebuttal by Authors (cont.)**
>
> > If the stream is in turnstile model, then the author should include comparison with Count Sketch (Charikar, Moses, Kevin Chen, and Martin Farach-Colton. "Finding frequent items in data streams.").
>
> In section A.3.4 of the Appendix, we include additional results comparing the Partitioned Learned Count-Sketch (PL-CS) to our PL-CMS on the Zipfian datasets. In all the experiments we use the same parameters for PL-CS as for PL-CMS using the actual $E_i$ and $F_i$ values, since we do not know how to optimize its parameters to minimize the FPR. We also apply the same best cutoff thresholds. We observe that the standard and learned CS of Hsu et al. (denoted as CS and LCS, respectively) perform similarly to basic CMS and LCMS. For the Macbeth dataset with $n/k=16$, PL-CS with 3 Count-Sketches (denoted as 3CS-A) achieves a worse FPR than the baseline algorithms and 3-CMS-A. For Bible with $n/k=879$, 2CS-A also underperforms and yields a much higher FPR than the baseline algorithms and 2CMS-A. It is also important to mention that using Count-Sketch induces a small false negative rate in the task of identifying the frequent elements which is not present for Count-Min Sketch variants. While we do not expect the optimal PL-CS to provide significant advantages over PL-CMS, optimizing its parameters is an interesting future research direction.

---

> > ### Comment · Reviewer_4TXw · 2023-11-22
> >
> > What is the challenge to use the proposed framework in optimizing count sketch?
> > Perhaps it is possible to use Chebyshev's inequality for equation (2) in analyzing count sketch and then union bound?

---

> ### Author Response · Authors · 2023-11-22
>
> Thank you for the suggestion, we believe your approach could yield an optimizable FPR formula. Using Chebyshev's, for a single hash function we would have $Pr(|\hat{f}_x - f_x|\geq \frac{\epsilon n}{k}) \leq\frac{ k \|f\|_2^2}{m_i\epsilon n}$. Union bounding over $t_i$ hash functions gives error probability $\leq \frac{ t_ik \|f\|_2^2}{m_i\epsilon n}$, so our false positive formula becomes $\text{FPR}\leq \sum_i^c F_i \cdot\frac{ t_ik \|f\|_2^2}{m_i\epsilon n}$. Given total space $S$, setting $r_i\cdot S = m_i\cdot t_i$ ,  we get $\text{FPR}\leq \sum_i^c F_i \cdot\frac{ t_i^2k \|f\|_2^2}{r_i S\epsilon n}$. Since $t_i=0$ is a minimizer but is not valid, we could potentially set it as a constant $1$.
>
> That would give the following optimization problem $\min_{r_1,\ldots r_c} \sum_i^c F_i \cdot\frac{k\|f\|_2^2}{r_i S \epsilon n}$ subject to $\sum_i^c r_i = 1$ and $r_1, \ldots, r_c \geq 0$. Ignoring the second constraint, this can most likely be solved using Lagrange multipliers yielding a closed-form solution.
>
> However, since Count-Sketch takes the median of the $t$ estimates, this upper bound is too loose to be useful. Due to using the Union bound, the FPR increases with larger $t$. Using the Median trick would give a tighter upper bound, but also a more complex FPR upper bound formula to optimize. Nonetheless, it is an interesting next step to explore. It is important to note, that the final algorithm would still need to estimate the $F_i$ quantities.

---

> > ### Comment · Reviewer_4TXw · 2023-11-23
> >
> > That makes sense. Thanks for the detailed response.
> > Perhaps see https://arxiv.org/pdf/1207.5200.pdf for the median of median trick for Count Sketch.
> > I think this framework has great potential.

---

### Official Review · Reviewer_jB1e · 2023-11-01

**Soundness:** 3 good
**Presentation:** 3 good
**Contribution:** 2 fair
**Rating:** 5
**Confidence:** 4

**Summary:**

The paper studies the frequency estimation problem in the learning-based setting where we aim to improve the performance of algorithms with the help of machine learning prediction. In the previous work of (Hsu et al. 2019), the main idea is to use the machine learning method to participate items into two sets. Items with sufficiently high predicted frequencies have their frequencies tracked exactly, while the remaining items, with low predicted frequencies, are placed into the Count-Min Sketch data structure. In this work, the authors extend this idea and propose the partitioned learned count-min sketch(PL-CMS) where the algorithm partitions the items into multiple ranges based on the prediction. The paper studies how to set the threshold of each range to make the performance of the algorithm better formally (in this paper,  the estimation error metric is different where the authors aim to improve the false positive rate of identifying the heavy items). The experiments also show the advantages of the proposed algorithm.

**Strengths:**

1. The theoretical contribution of the paper is solid. The idea of extending the prediction to multiple ranges is natural. However, the analysis of how to set the thresholds of each range is not clear. The paper gives a formal analysis of this.

2. The presentation of the paper is clear and easy to follow.

**Weaknesses:**

1. The paper does not give a study of the cases when the machine learning prediction is noisy, which is one of the central parts of the previous works. In this work, we want to partition the items into multiple ranges, hence the requirement of the prediction precisions is even higher and the study of the algorithm using the noisy prediction is even more important.
(one related model in [1] is rather than predict the range each item will be in, we instead assume the prediction can give an approximation of the frequency of each item. with an alpha additive error and beta multiplicative error)

[1] Justin Chen et al. Triangle and Four-Cycle Counting with Predictions in Graph Stream. ICLR 2022

**Questions:**

1. In this paper, the definition of the heavy items we are interested in is the i such that $f_i \ge n/k$. In a number of the works, the heavy hitter also be defined as $f_i \ge \sum_j f_j / k$, can the analysis in this work be extended to this model?

2. In the experiments, the authors study the performance of the algorithm using both the ideal prediction and the noisy prediction. The result shows that there are still some gaps in performance between the two cases. I think it would be an interesting part if the author could give an (brief) analysis of the precision of the current prediction.

---

> ### Author Response · Authors · 2023-11-18
> **Rebuttal by Authors**
>
> Thank you for your time and helpful comments. We address the questions and concerns below:
> >The paper does not give a study of the cases when the machine learning prediction is noisy, which is one of the central parts of the previous works. In this work, we want to partition the items into multiple ranges, hence the requirement of the prediction precisions is even higher and the study of the algorithm using the noisy prediction is even more important. (one related model in [1] is rather than predict the range each item will be in, we instead assume the prediction can give an approximation of the frequency of each item. with an alpha additive error and beta multiplicative error)
>
> Given fixed thresholds, even when the predictions are noisy or completely inaccurate, PL-CMS optimizes the space allocations of each CMS accordingly, which means that PLCMS is naturally robust. Specifically, we use the predictions to estimate the quantities $E_i$ and $F_i$ as well as the number of unique buckets from the training dataset. If, for example, we incorrectly place a large number of infrequent items in the first region, $F_1$ will be large and we will assign $\text{CMS}_1$ a lot of space to lower its FPR.
>
> As an example -- consider the case when our ML model is just a random oracle that uniformly assigns elements of the stream into different regions of PL-CMS, without any correlation to their actual frequencies. Further, assume that the number of unique buckets is small (which is typically true in practice). In this setting, one can show that the expected error of any item under PL-CMS is roughly equal to its expected error under the standard Count-Min sketch -- that is, when the learned model degenerates to random guesses, PL-CMS still matches the performance of the unlearned algorithm, indicating a high level of robustness. The argument here is just that, since a random oracle will allot every region a roughly equal fraction of infrequent items $F_i$ and a roughly equal total frequency of items $E_i$ and, as a result, each CMS will be assigned the roughly same amount of buckets $m_i$, we will have $\frac{E_i}{m_i}\approx \frac{n/c}{m/c} = \frac{n}{m}$, where $c$ is the number of regions. This is the same ratio (and in turn yields the same false positive rate) as if we assigned all items to a single CMS with $m$ buckets. Note that a similar argument would hold if the model assigned items to different regions with different probabilities. In this case, letting $p_i$ be the probability of an item being assigned to region $i$, we would have $m_i \approx m \cdot p_i$.
>
> To verify the above argument, we have included additional experiments in section A.3.3 of the Appendix, where we show that given a learned model which outputs random predictions, 2-CMS and 3-CMS perform similarly to standard CMS and LCMS on the Zipfian datasets except for 3-CMS on the Macbeth dataset with $n/k=16$ which we are currently investigating.\\
>  We agree that establishing a measure of prediction error such as an alpha additive error and beta multiplicative error would be a very interesting future direction and could potentially help us optimize the partition thresholds.
>
> > In this paper, the definition of the heavy items we are interested in is the $i$ such that $f_i\geq n/k$. In a number of the works, the heavy hitter also be defined as $f_i\geq \sum_j f_j/k$, can the analysis in this work be extended to this model?
>
> We define $n$ as the sum of all frequencies in the data stream so the definitions are equivalent. The number of distinct items in the stream does not directly come into our bounds.
>
> > In the experiments, the authors study the performance of the algorithm using both the ideal prediction and the noisy prediction. The result shows that there are still some gaps in performance between the two cases. I think it would be an interesting part if the author could give a (brief) analysis of the precision of the current prediction.
>
> We believe the reviewer asks about the effect of estimating the $E_i$, and $F_i$ values on the algorithm's performance. We agree that it is an interesting direction to pursue. While we currently do not have a formal analysis of the estimates' precision, since we assume that the training dataset comes from the same distribution as the testing dataset, we expect the approximated $E_i$ and $F_i$ to be close to the true values (which we observe in practice).

---

> > ### Comment · Reviewer_jB1e · 2023-11-22
> > **Response to the Authors**
> >
> > Thanks for the detailed response. The authors show that even if the prediction is completely inaccurate (like a random oracle), the performance of the proposed approach will still be no worse than the standard approach. I think it will make the submission stronger if the author can make it into the main body of the paper.
> >
> > However, in my opinion, in addition to the worst-case guarantee, another important and interesting question here is under which condition, the noisy prediction will still make the algorithm have a strictly better performance than the standard approach? Or can the error be related to some quantity that measures how noisy the prediction is? Hence, my opinion is the current version of the paper is on the borderline of acceptance.

---

### Official Review · Reviewer_PmnJ · 2023-11-09

**Soundness:** 3 good
**Presentation:** 4 excellent
**Contribution:** 2 fair
**Rating:** 6
**Confidence:** 4

**Summary:**

In their paper, the authors introduce a novel approach for efficient heavy hitter frequency estimation, referred to as PL-CMS, which leverages a learned Count Min Sketch (CMS) technique across multiple score partitions generated from a trained model. This method builds upon prior research, notably the 'Learned Count Min Sketch' (LCMS), which employs a single score threshold, as well as the work of Dai and Shrivastava (2020) and Vaidya et al. (2020), where multiple partitions are utilized for a learned Bloom filter.

The key advantage of PL-CMS is its ability to achieve lower false positive rates while adhering to specific space constraints. The authors demonstrate the effectiveness of their approach through experiments conducted on four real-world datasets.

**Strengths:**

1. PL-CMS performs better with lower False Positive Rates compared to LCMS and CMS.
2. The approach although derived and inspired from the existing works, fills are right gap in the literature of learned CMS structures.
3. The theoretical analysis provided an upper bound on False positive rate.

**Weaknesses:**

1. The solution is a simple extension of Dai and Shrivastava (2020) and Vaidya et.al. (2020) for LCMS. In my opinion it discounts the novelty. However it is not a strong criticism against the paper.
2. Fig 2 legends will help.
3. The choice of parameters are not well explained (page 7 para 4 and Section 4.5). How does that relate to Fig 2?
4. page 7 para 4- “We ignore the space of the learned model itself, which is lower order is our settings”. Please provide space taken by model and CMS tables together for each dataset to justify the statement.

**Questions:**

1. Comparison with other 2 methods of Zhang et. al. (2020): The paper hinted that their approach involves highly accurate learned model and hence omitted for comparison. Are there ways to compare them on equal grounds?
2. What is the model size for LCMS and PL-CMS for each datasets? Can we train them to achieve similar accuracy levels as in Zhang et. al. (2020)? What are the bottlenecks?
3. Is the code available for replication of plots/results?

---

> ### Author Response · Authors · 2023-11-18
> **Rebuttal by Authors**
>
> Thank you for your positive review and helpful comments. We address the main questions and concerns below:
> > The choice of parameters are not well explained (page 7 para 4 and Section 4.5). How does that relate to Fig 2?
>
> We choose the thresholds by performing a grid search on the training dataset and picking thresholds that yield the lowest average FPR in identifying $(\epsilon,k)$-frequent items in a wide range of space allocations. Assuming the length of the testing dataset $n$, we scale the found thresholds by $n/n_{tr}$ where $n_{tr}$ is the length of the training dataset. Specifically, to find the first threshold $l_1$, we incrementally iterate through a wide range of cutoff values and report the one that yields the lowest FPR for LCMS on the training dataset. Next, we fix $l_1$ and perform another grid search to find the next cutoff $l_2< l_1$ which yields the lowest FPR for 2-CMS. Similarly, to find $l_3$ (where $l_3 < l_2$), we fix $l_1$ and $l_2$. We let $l_3$ be the cutoff value which yields the lowest FPR for 3-CMS using $l_1$,$l_2$ and $l_3$. In the final step, we rescale the thresholds by $n/n_{tr}$. We will include a more detailed description of the process in our paper.
>
> Figure 2 explains the motivation behind introducing the parameter $p$ in our theoretical false positive rate upper bound to make the bound more closely reflect the false positive rates observed in practice. We use the updated formula to optimize the space allocations of each region and its parameters given fixed thresholds found from the grid search. Note, that the thresholds determine the $E_i$ and $F_i$ parameters in each region used to optimize the space allocations.
>
> > Comparison with other 2 methods of Zhang et. al. (2020): The paper hinted that their approach involves highly accurate learned model and hence omitted for comparison. Are there ways to compare them on equal grounds?
>
> We were not able to compare the two algorithms on equal grounds on the Beijing PM2.5 dataset. We trained a learned model according to the approach of Zhang et al. with the same architecture and specified $5\%$ training dataset split size as well as the reported parameters but our resulting model was not accurate and had an MSE of $31552.73$. Since the authors' algorithm relies on accurate predictions and they did not describe how to optimize its parameters, we were not able to proceed further. We also tried to run Zhang's algorithm using our model's architecture and the same train-test ratio of 8:2. For $n/k=420$ with space allocation of $1954$ units, Zhang's algorithm (with parameters reported in their paper) yields an FPR of $0.9$, whereas PLCMS achieves a much lower FPR of $0.367$ for the same space budget.
>
> > What is the model size for LCMS and PL-CMS for each datasets? Can we train them to achieve similar accuracy levels as in Zhang et. al. (2020)? What are the bottlenecks?
>
> We use the same learned model for LCMS and PL-CMS. The size of the model for the Zipfian datasets is equal to the size of the dictionary that stores the rank of each word in the training dataset. Following the approach of Hsu et al., we focus on the space complexity of the data structure instead of its learned model which can be amortized over time (i.e., if identifying frequent items over many days or other time periods, the model remains fixed, while a new data structure is used in each round).
>
> Since the approach of Zhang et al. requires adjusting the frequencies of items during training, it cannot be directly applied to the Zipfian datasets. Furthermore, Zhang's algorithm requires retraining the model for every different $n/k$ value as well as fine-tuning many of its parameters.
>
> > Is the code available for replication of plots/results?
>
> While the code is not currently published, we are working to make it available.

---

### Meta-Review · Area_Chair_Tqbr · 2023-12-04

**Metareview:**

This paper builds on recent work on leaded frequency item sketches.  The paper presents a partitioned approach,   known algorithms in the partitions.  The reviewers generally liked the idea of a partitioned approach to count sketch and this could be a promising idea.  A shortcoming pointed out by several reviewers is that an analysis of the performance trade-off with error in the predicted parameter is lacking.  More theoretical and empirical justification into what happens with a noisy predictor would have made the paper stronger.

**Justification For Why Not Higher Score:**

The main shortcomings are (1) the paper did not have a champion.  (2) I, like the reviewers, would have liked to see more emphasis on the robustness of the algorithm proposed when prediction errors are present.

**Justification For Why Not Lower Score:**

N/A

---

### Decision · Program_Chairs · 2024-01-16

Reject